# Suggested Research Trends in the Area of Micro-EDM—Study of Some Parameters Affecting Micro-EDM

**DOI:** 10.3390/mi12101184

**Published:** 2021-09-29

**Authors:** Atanas Ivanov, Abhishek Lahiri, Venelin Baldzhiev, Anna Trych-Wildner

**Affiliations:** 1Department of Chemical Engineering, Brunel University London, London UB8 3PH, UK; venelin.baldzhiev@brunel.ac.uk; 2Institute of Metrology and Biomedical Engineering, University of Warsaw, 02-093 Warsaw, Poland; venelinbaldzhiev@yahoo.co.uk

**Keywords:** micro-EDM, tool electrode materials, micro-EDM process parameters

## Abstract

This paper provides an overall view of the current research in micro-electrical discharge machining (micro-EDM or µEDM) and looks into the present understanding of the material removing mechanism and the common approach for electrode material selection and its limitations. Based on experimental data, the authors present an analysis of different materials’ properties which have an influence on the electrodes’ wear ratio and energy distribution during the spark. The experiments performed in micro-EDM conditions reveal that properties such as *electron work function* and *electrical resistivity* strongly correlate with the discharge energy ratio. The electrode wear ratio, on the other hand, is strongly influenced by the *atomic bonding energy* and was found to be related to the tensile modulus. The proposed correlation functions characterized the data with a high determination coefficient exceeding 99%.

## 1. Introduction

Non-conventional processes such as electrochemical machining, electrical discharge machining, and laser machining are nowadays widely present in research as well as in industrial applications [1,2]. They are used in the production of home appliances, personal well-being products [3], aviation components [4], medical instruments, and automotive parts [5,6]. Despite the presence of these technologies, the rapid development of the consumer market and the competition for achieving better product quality forces the search for new concepts and solutions, even in well-established manufacturing processes. Limitations concerning in-depth knowledge of non-conventional micro-machining and especially the lack of detailed theoretical understanding of the physical phenomena describing the material-removal process are major obstacles in achieving better control on micro and nano scales.

Recent efforts in the research and industrial applications of micro-EDM are of great interest as they promise to satisfy, to a great extent, the constant desire for miniaturization [7,8]. Other areas of interest are hybrid processes, deploying at least two different techniques [9], and even nano-scale machining [10,11]. Unfortunately, the majority of the research community devotes its attention to specific commercial applications and, in this way, misses the global scientific approach and scientific exploration of the processes’ behavior. However, the micro-processes still need to be investigated and developed to face new challenges, as they are not fully understood, and with newly acquired knowledge to bring novel possibilities. This paper will attempt to present the scientific side and offer trends in the further research of the micro-EDM process.

## 2. Micro-EDM

In general, micro-fabrication or micro-manufacturing can be defined as the fabrication of products or components where the dimensions of at least one feature are in the micrometer range. This is a common definition of micro-manufacturing as proposed in [12], which has also been adopted by others to classify the manufacturing processes [13,14,15]. This, however, cannot be the only criterion for the classification of the machining process, especially in terms of micro-EDM.

Micro-electrical discharge machining is one of the non-conventional manufacturing technologies that are increasingly present in the production of micro molds and dies [16,17] and many other commercial products [3]. In micro-EDM, both tool and workpiece are scaled down to enable manufacturing of microfeatures, ranging from a few *mm* to several *µm* [18] or even several *nm* [10,11], so, in conventional terms, the difference between EDM and micro-EDM processes is mostly connected with downscaling the process. However, it is important to notice that even though the discharge energy of a single pulse is typically in the range of *µJ* [19,20] or even *nJ* [21], the energy density usually goes up, and as shown in [22], the main difference between macro- and micro-EDM is the *energy density* acting on the surface. Therefore, the big question here is: by scaling all dimensions down, do the processing parameters stay in the normal boundaries for EDM, or do they change drastically, which in turn brings changes in the mechanism of material removal? 

Such a “scale down” approach can only be justified if it takes into account the duration of the discharge pulses in macro- and micro-EDM and their effects on the workpiece surface. Hence, a better distinction between these two processes can be clarified if the energy densities are compared.

In this paper, it is assumed that the crucial difference between macro- and micro-EDM is the plasma channel formed during the discharge. In macro-EDM, the size of the tool electrode is much bigger than the radius of the plasma channel, and pulses are of much longer duration. The situation in micro-EDM is different. In general, the radius of the plasma channel is determined by the pulse duration. The longer the pulse, the more the plasma channel expands, and the larger the affected surface area becomes. In micro-EDM, the discharge pulses are much shorter, and that does not allow the plasma channel to expand. However, the size of the tool is also much smaller. This leads to a greater energy density affecting the workpiece surface, and the assumption is that different phenomena of material removal in micro-EDM appear, which leads directly to material ablation rather than melting vaporization [23]. Changes in the mechanism of material removal in EDM when very short pulses are used have been reported by manufacturers of micro-EDM machines such as SARIX. They reported that when machining with very short pulses, the recast layer was largely reduced or not present at all.

This assumption is consistent with the size and shape of produced microcraters due to the absence of material melting during the process. The “erosion efficiency” of the micro-EDM process is proportional to the ratio of actual erosion energy vs. supply energy, which is much higher for lower energies applied using RC-pulse discharges compared to when higher energies are applied. It was also noticed that lower-energy discharges produced a more consistent size of microcraters at higher efficiency, which was reported in [19].

Therefore, the EDM process on the micro-scale can be described not as the miniature version of the macro-EDM process but as a process in which the physical phenomena are different. That difference is due to shorter discharge pulses which affect the surfaces with much higher energy density. Such an approach contradicts the classical definition of the micro process presented in [12], which perceives the processes in terms of dimensions of tools and manufactured products. This also means that with very short pulses, larger tool electrodes can also effectively perform the micro-EDM process that authors in [24] demonstrated using larger electrodes (1.5 mm) in micro-EDM. Therefore, the main question now is: what is the scientific basis for these phenomena, and what are the boundaries? It seems logical that for different electrode materials, these new phenomena may have different boundaries.

## 3. Typical Electrode Material Selection Approach

One major challenge for micro-EDM is the fabrication of tool electrodes. Manufacturing them and monitoring the results to achieve dimensions in the expected range tends to be a separate research topic [25,26,27].

A key issue in micro-EDM as well as in conventional EDM is the selection of a suitable material for the tool electrodes. This selection is crucial because of the need for high electric conductivity of the material used, which is essential, and good machinability to allow for easy forming of the appropriate electrode shapes, especially for micro-EDM. Recent experiments with non-conductive electrodes are being performed [28,29,30], but in general, the typical tool materials observed in EDM are graphite, copper—due to good machinability and thermal and electric conductivity—and tungsten, due to high melting temperature. Similarly, these are being applied in micro-EDM [31], together with tungsten carbide and brass [32] or other materials based on those—for example, alloys such as cupronickel [33]. There are also other, less popular, materials used as electrodes to verify their possible application in the micro-EDM process, such as cermet [34], composites [35], materials based on polycrystalline diamond and boron-doped CVD-diamond [36], materials with additional coatings [37], and even carbon fibers [24,38].

The workpiece materials are determined mainly by the application and functionality of the end product. One of the typical materials, machined by EDM and used for many applications, is stainless steel (304, 316) [39,40,41]. Other materials worth mentioning are titanium and its alloys [42,43,44], composite materials [15,45], and difficult-to-machine alloys such as Inconel [46,47,48].

In most cases, the selection of the electrode materials is to optimize machining parameters and to achieve certain values of process efficiency, measured by attaining the design parameters of the product, material removal rate (MRR), and tool wear rate (TWR), which is the ratio of the loss of material from the two electrodes [46,48,49]. The other approach to material selection is the analysis of process performance and identification of key factors influencing the results [25,33,36]. Such an approach establishes the processing window in which the manufacturing should be performed, the crucial input factors that will affect the process significantly, and the relationships linking them with outputs such as MRR, TWR, surface roughness, etc.

## 4. New Electrode Material Selection Approach

Many materials are selected for electrodes based on optimization procedures or laboriously planned experiments [25,49]. Such an approach is common because it gives detailed characteristics of the process with newly tested materials [33,36,41,46,48,49]. However, it is usually limited to a certain selected material(s) of the tool/workpiece pair chosen for the experiments. Selecting materials for electrodes in this way can be regarded mainly as a case study that gathers data for specific materials and process behaviors. Although this approach is good for rapid testing of newly developed applications and for finding optimal parameters of the machining process [46,47,48], the findings from such tests may not be transferable to other materials. Thus, it is difficult to generalize about the workpiece or tool material under erosive conditions because specific types of materials, even those sharing some similarities, have a large variety of different physicochemical properties, chemical compositions, and metallographic structures that can affect the process results. At present, there is a lack of a scientific approach able to predict material behavior in the EDM process.

The change in physicochemical properties of designed material can lead to different process results, as researchers showed in [50]. When analyzing different materials for tool electrodes, their thermal properties—such as boiling and melting point and thermal conductivity—should be carefully considered because EDM is mainly a thermal process. However, should all these material properties be considered? As assumed above, if the pulse discharge is very short, ablation of material takes place without heat transfer.

In order to go in-depth on a purely theoretical basis and not be linked to any practical application, one paper’s experiments investigated the behavior of pure materials with a purity greater than 99.95% [20]. The experiment results showed that electrode materials have a significant effect on the process’s electrical parameters, such as energy distribution and current. Therefore, the selection of the tool electrode material is dependent on the workpiece material, and it is possible to design material(s) with specific properties depending on the properties of the workpiece material. 

## 5. Energy Distribution Analysis

Taking into account the above-mentioned consideration about the micro-EDM process, the following analysis can be performed. The energy during the discharge is distributed between cathode, anode, and the dielectric medium as presented in Figure 1 and denoted in Equation (1).
(1)$$E_{dis}=E_{dis}^{anode}+E_{dis}^{cathode}+E_{dis}^{dielectric}$$

For the sake of this analysis, it is assumed that in micro-EDM, the volume of the material removed ($V_{dis}$) from the workpiece and from the tool electrode is proportional to the discharge energy (2). At this point, it is not known if *K* is a constant or a function of the discharge potential and material.
(2)$$V_{dis}=KE_{dis}$$

Typically, and well-accepted is that the energy in the EDM process can be described according to Figure 2, where: *t*_0_—time off, *t*_i_—time on, *t*_e_—time of the discharge, *i*_max_—max current, *i*_dis_—discharge current, *V*_0_—supply voltage, *V*_e_—discharge voltage. 

Analyzing Figure 2, Equations (3) and (4) for the energy can be written as:(3)$$E_{dis}=\int_{0}^{t_{e}}W_{t}dt$$
where the power *W* (time-dependent) is:(4)$$W_{\left(t\right)}=I_{dis}\left(t\right)V_{e}\left(t\right)$$
hence the energy can be expressed as:(5)$$E_{dis}=\int_{0}^{t_{e}}I_{dis}\left(t\right)\bar{V_{e}}\left(t\right)dt$$

The initial assumption was that in micro-EDM, the material is directly vaporized, and the following modeling does not include the thermal conductivity of the material. The suggested modeling does not pretend to be absolute truth, but at the same time, it becomes valuable for further investigation and direction of the research. It is considered that the enthalpy of the system (*H*) can be used to describe the internal energy (*E*) of the system and EDM process according to Equation (6), where P and V are pressure and volume, respectively:(6)H=E+PV

As the EDM process is complex, the two main contributions to energy can be identified. One is connected with the melting process and the other with the vaporization process, (7) and (8), respectively:(7)$$E_{m}=mH_{m}$$
(8)$$E_{v}=mH_{v}$$
where *m* is the mass, *V* is the volume, and *ρ* is the density expressed in Formula (9)
(9)m=Vρ

Thus, the energy to vaporize material from both cathode and anode can be described by Expressions (10) and (11)
(10)$$H_{vap}^{anode}=\int_{T_{0}}^{T_{m}}c_{p}^{anode}dT+L_{m}^{anode}+\int_{T_{m}}^{T_{v}}c_{p}^{anode}dT+L_{v}^{anode}$$
(11)$$H_{vap}^{cathode}=\int_{T_{0}}^{T_{m}}c_{p}^{cathode}dT+L_{m}^{cathode}+\int_{T_{m}}^{T_{v}}c_{p}^{cathode}dT+L_{v}^{cathode}\;$$
where *c*_p_, *L*_m_, *L*_v_, *T*_0_, *T*_m_, and *T*_v_ are, respectively, the specific heat capacity, latent heat of melting, latent heat of vaporization, ambient temperature, melting temperature, and vaporization temperature.

Considering the micro-EDM process with component form vaporization, as it was mentioned above (in Section 2, paragraph 4), the following distribution of energy can be obtained. For Equation (8), the energy for both electrodes is expressed in formulae (12) and (13).
(12)$$E_{vap}^{anode}=V_{vap}^{anode}\rho_{anode}H_{vap}^{anode}$$
(13)$$E_{vap}^{cathode}=V_{vap}^{cathode}\rho_{anode}H_{vap}^{cathode}$$

Further, considering the electrode wear ratio, which is the ratio of the lost volumes of both electrodes and taking into account (12) and (13) to represent the vaporized volumes $V_{vap}^{anode}$ and $V_{vap}^{cathode}$, the following equation is obtained:(14)$$\nu =\frac{V^{anode}}{V^{cathode}}=\frac{E^{anode}\rho^{cathode}H_{vap}^{cathode}}{E^{cathode}\rho^{anode}H_{vap}^{anode}}$$

This equation brought the idea to use the same material for anode and cathode. If the materials for both electrodes are the same, then the wear ratio will only depend on the energy distribution between the anode and cathode (15). Now comes the big question: is this ratio constant, or is it somehow material-dependent?
(15)$$\nu =\frac{E^{anode}}{E^{cathode}}$$

The above-mentioned analysis reduced the wear ratio only for the energy provided to both electrodes. The detailed experiment planning and results of eroding pure metals against each other are given in [20]. The authors found that depending on the material, the discharge energy and other electrical parameters are different, providing the same sparking parameters in each test, and the results can be observed in Figure 3.

This leads to the conclusion that some material properties, when both electrodes are of the same material and experiencing the same sparking conditions (parameters), influence the energy distribution and hence the electrode wear ratio.

## 6. Analysis of Possible Material Properties Influencing the Energy Distribution during the Micro-EDM Process

The aim of the experiments was to find the properties that affect the energy distribution during the micro-EDM process. In the tests, the electrodes were of pure metals, presenting a wide range of physicochemical properties. The list included Ti, W, Ag, and Ni but also the metal typically used for tool electrode material—Cu. All electrodes in the experiments were wire (rod) with a 1 mm diameter. The parameters used in the experiments were the same for all the tests and are summarized in Table 1. More than 2600 randomized tests were completed in order to avoid the effect of any systematic factors on the results.

The methodology of the analysis first assumed the calculation of the correlation coefficient between the properties of each examined material, single discharge energy, and wear ratio, respectively. Then, properties showing high correlation values were chosen to obtain the fitting formula. 

## 7. Estimation of the Energy Distribution Functions Based on Material Properties

To estimate the dependency between the results of the experiments and material properties, the correlation coefficient between measured impulse energy and various material properties has been calculated. Table 2 presents selected properties for all analyzed materials with their correlation coefficients regarding energy distribution. A strong correlation with the energy is observed in the case of electron work function and electrical resistivity. 

The electron work function exhibits a much higher correlation coefficient than the key factor for the material to be considered suitable for electrodes both in EDM and micro-EDM—electrical resistivity [44]. The value is close to the total positive linear correlation (equal to 100%). As the electron work function represents the energy that must be supplied to an electron to cross over the surface barrier of a metal, it indicates that it can be expected that materials with a high value of this property will lead to higher measured energy in the same process conditions.

Table 3 below shows the data and the units for the plot on Figure 4 which is the linear relation between the electron work function and the share of the energy depending on the material. 

The determination coefficients for the electron work function were 99.345% and 99.348% for linear and second-degree polynomial fittings, respectively. For the electrical resistivity, they were 62.11% and 71.51%, respectively.

By combining both the most important factors for the energy distribution during sparking, *R*^2^ = 99.9993808 can be achieved. It has to be mentioned that the electron function for one material can vary depending on the structure of the material. This will allow, when knowing the workpiece material, to design the electrode material and predict the energy distribution between the two electrodes. This will make the micro-EDM process more predictable.

## 8. Estimation of the Wear Ratio Distribution Functions Based on the Properties of Materials

To estimate the dependency between the results of the experiments and selected material properties, the correlation coefficient between the wear ratio and various material properties was calculated. Table 4 presents these material properties with their correlation coefficient in regard to the wear ratio. The wear ratio exhibits a strong correlation with the atomic bonding energy (represented here as tensile modulus), thermal conductivity, and coefficient of thermal expansion.

Table 5 gives the data of the four material properties most closely correlated to the wear ratio of the electrode. Most important is the atomic bonding energy (for convenience, it is represented here by tensile modulus).

The determination coefficients for the tensile modulus were 72.37% and 93.84 % for linear and second-degree polynomial fittings, respectively, and the plot is given in Figure 5. For thermal conductivity, the coefficients were 54.48% and 72.71%, respectively, and for thermal expansion, – 53.00% and 55.34%, respectively.

Although the melting point from Table 4 has a similarly high correlation coefficient, it was not considered in this study. The melting point is important in terms of the evaluation of wear ratio [44]; however, as the analyses were made for micro-EDM, in which evaporating of the materials due to high energy density rather than melting is dominant, the melting point was not included in this study.

## 9. Conclusions 

The present work aims to bring new theoretical insights into existing research on the micro-EDM process and to try and open new horizons for future research and discussions. Based on the analysis, it appears that there is a linear relationship between the work function of the material and measured energy. Additionally, a parabolic relation between the electrode wear ratio during micro-EDM and the tensile modulus of the electrode material is observed. These relationships indicate that one can approach ED micromachining of the materials from a theoretical perspective and apply relevant and optimum energy for successful EDM. 

## 10. Future Prospects

Future research must be conducted in order to establish the boundaries of each of the mechanisms of material removal, which would allow reliable prediction of process behavior:Build up a theoretical model, which is then proven experimentally, to be used for establishing the energy density needed to directly vaporized material during the EDM process. Obviously, this will be an important parameter to mark the change of the material-removal mechanism and the disappearance of the recast layer.The suggested model using enthalpy must be improved to not follow the melting–vaporization stages but go directly through ablation (sublimation), as shown in Figure 6.More accurate models of the energy distribution during sparking must be developed, which will allow knowing the specific material properties of the workpiece in order to select or even to design electrode materials, and in this way to control the energy distribution during sparking. This, in turn, will dramatically reduce the electrode wear and improve process predictability.Based on the presented research, a new model for electrode wear must be developed, keeping in mind the material-removal mechanism and material-bonding energy.Electrode material(s) can be selected (designed) to have much less electrode wear with high predictability of the outcome.The micro-EDM process has to undergo a high-level scientific investigation to make it more predictable and reliable. It is very likely, as well, that new parameters will be defined, and new micro-EDM machines will be designed and manufactured.

## Figures and Tables

**Figure 1 micromachines-12-01184-f001:**
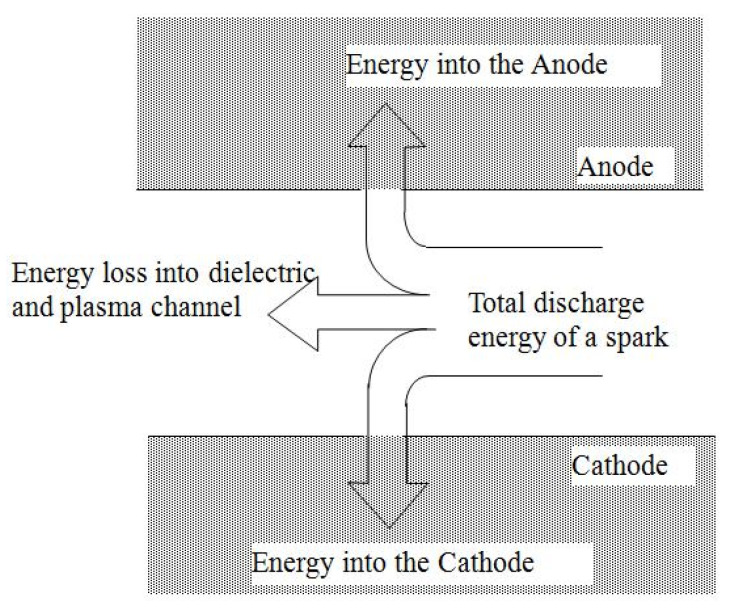
Energy distribution during sparking.

**Figure 2 micromachines-12-01184-f002:**
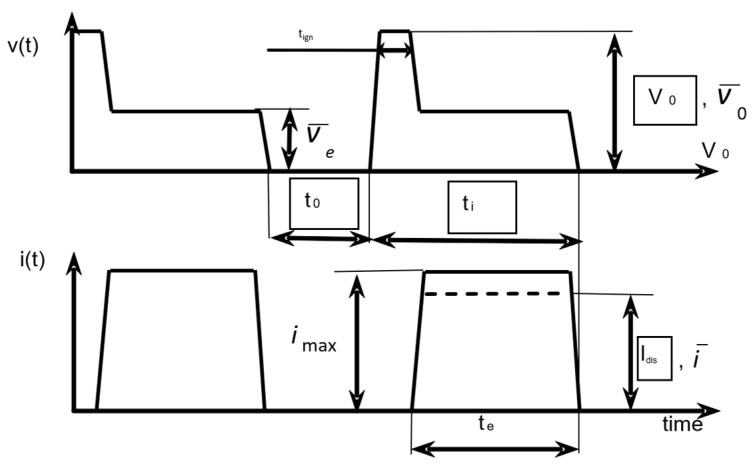
Typical waveforms during the discharges.

**Figure 3 micromachines-12-01184-f003:**
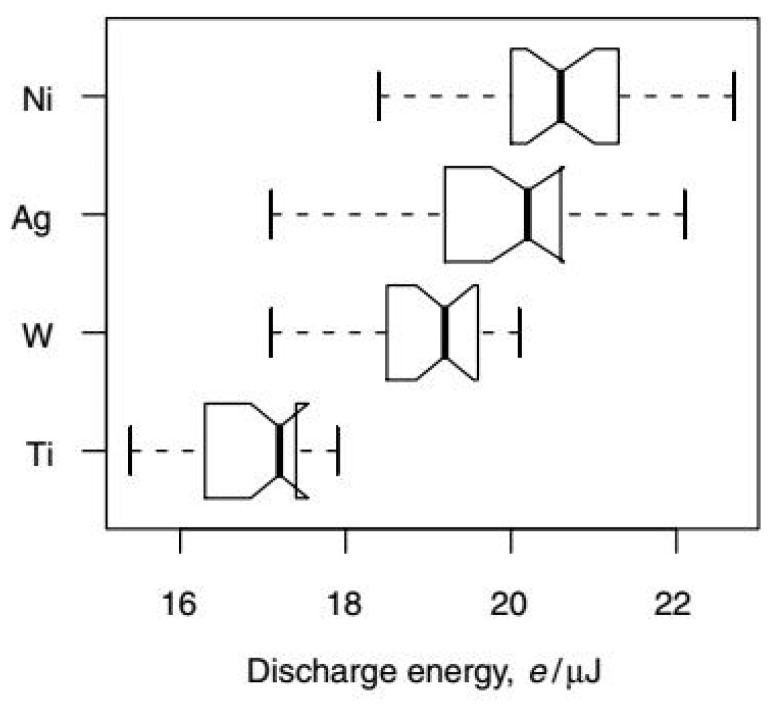
Discharge energy for different materials [20].

**Figure 4 micromachines-12-01184-f004:**
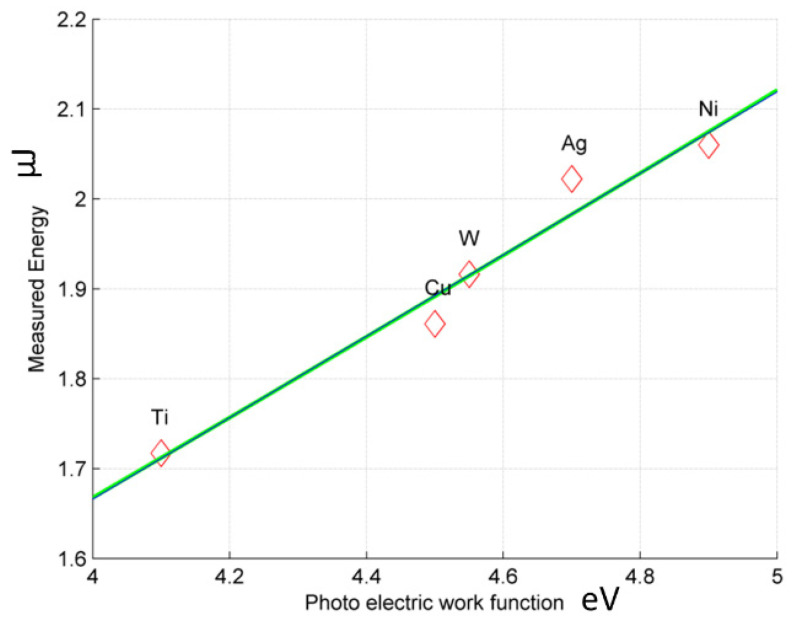
Energy against electron work function.

**Figure 5 micromachines-12-01184-f005:**
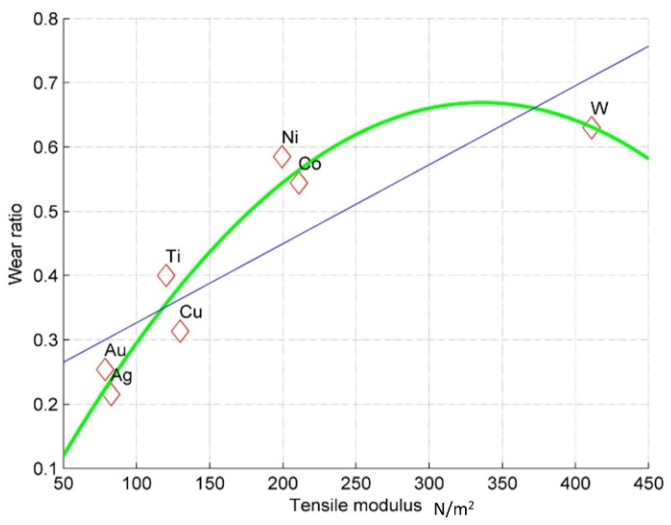
Wear ratio against tensile modulus.

**Figure 6 micromachines-12-01184-f006:**
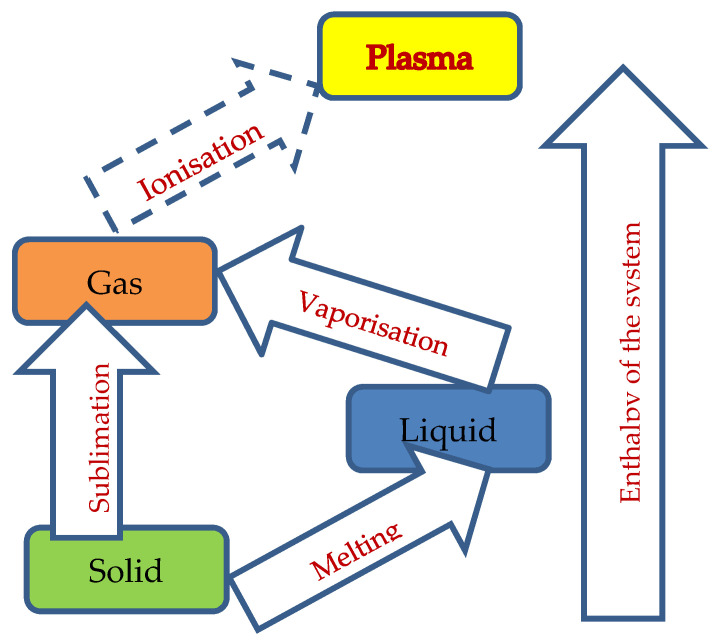
Routes for material vaporization by increasing the enthalpy of the system.

**Table 1 micromachines-12-01184-t001:** Parameters in experiment.

Parameter, Symbol	Unit, Symbol	Value
Open circuit voltage, *V*_0_	Volt, *V*	80
Average current from the generator, *I*	Ampere, *A*	0.5
Duration of the pulse of voltage at *V*_0_, *T*-on	Microsecond, *μs*	1
Programmed time interval between adjacent pulses of voltage, *T*-off	Microsecond, *μs*	1
Reference voltage of the servo system^a^	Volts, *V*	50

**Table 2 micromachines-12-01184-t002:** Correlation coefficients of energy distribution for different material properties.

Material Property	Energy
Atomic number	0.28
Atomic radius	−0.46
Atomic weight	0.24
Electron work function	0.98
Thermal neutron absorption cross-section	0.44
Temperature coefficient	0.67
Electrical resistivity	−0.78
Boiling point	−0.21
Density	0.35
Melting point	−0.13
Coefficient of thermal expansion	0.40
Latent heat of evaporation	−0.62
Latent heat of fusion	−0.55
Specific heat	−0.40
Thermal conductivity	0.32
Bulk modulus	0.17
Tensile modulus	0.10
Enthalpy	−0.62
Enthalpy/thermal conductivity	−0.75

**Table 3 micromachines-12-01184-t003:** Data used in calculations and plotting of Figure 4.

Materials	Measured Energy (µJ)	Electron Work Function (eV)
Ti	1.717	4.1
Cu	1.861	4.5
W	1.916	4.55
Ag	2.022	4.7
Ni	2.06	4.9

**Table 4 micromachines-12-01184-t004:** Correlation coefficients of wear ratio for different properties.

Material Property	Wear Ratio
Atomic number	−0.12
Atomic radius	−0.45
Atomic weight	−0.11
Electron work function	0.18
Thermal neutron absorption cross-section	−0.57
Temperature coefficient	0.71
Electrical resistivity	0.06
Boiling point	0.60
Density	0.04
Melting point	0.73
Coefficient of thermal expansion	−0.73
Latent heat of evaporation	0.49
Latent heat of fusion	0.57
Specific heat	0.22
Thermal conductivity	−0.74
Bulk modulus	0.72
Atomic bonding energy (tensile modulus)	0.85
Enthalpy	0.48
Enthalpy/thermal conductivity	0.09

**Table 5 micromachines-12-01184-t005:** Calculated or derived data relevant for the wear ratio.

Materials	Measured Wear Ratio	Tensile Modulus (N/m^2^)	Thermal Conductivity (Wm^−1^K^−1^)	Thermal Expansion (K^−1^)
Ag	0.215	82.7	429	19.1
Au	0.254	78.5	318	14.1
Cu	0.3135	129.8	401	17
Ti	0.4	120.2	21.9	8.9
Co	0.544	211	100	12.5
Ni	0.585	199.5	90.9	13.3
W	0.63	411	173	4.5

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
