# Peer review of "Suggested Research Trends in the Area of Micro-EDM—Study of Some Parameters Affecting Micro-EDM"

_micromachines, 2021, doi:10.3390/mi12101184_

Round 1

Reviewer 1 Report

Dear author,

The paper "Suggested Research Trends in the Area of Micro-EDM—Study of Some Parameters Affecting Micro-EDM" is good and logical. And also, it is significance into the present research in micro-EDM area and describes an analysis of different materials’ properties that have influence on wear ratio and energy distribution during the spark. This paper has been reviewed but it needs major revision before accepted. The followings are the points need to modify.

  1. The first occurrence of the abbreviation must appear in the full name.
  2. The reference needs to be more recently, eg. 2020-2021.
  3. Also, the reference should be in order, not in random.
  4. The English must be revised. Some of the sentence or paragraph is really hard to read. They miss the Punctuation. Eg. The 4th para in page 2, there should have a comma after in this paper.
  5. The unit need to be shown in the axis.
  6. The conclusion need to be more specific and more detail results.
  7. The future prospects should contains some application of the EDM.

Author Response

Thank you for the review. Please, find below the addressing of the suggested corrections:

  1. The first occurrence of the abbreviation must appear in the full name.

             The manuscript has been edited accordingly.

  1. The reference needs to be more recently, eg. 2020-2021.

             Recent papers and research were studied and then added to the list of references.

  1. Also, the reference should be in order, not in random.

             The order of the references has been reorganised.

  1. The English must be revised. Some of the sentence or paragraph is really hard to read. They miss the Punctuation. Eg. The 4th para in page 2, there should have a comma after in this paper.

             The paper was edited for overall English, punctuation and spelling mistakes. 

  1. The unit need to be shown in the axis.

             The units in the graphs has been added in all the relevant figures.

  1. The conclusion needs to be more specific and more detail results.

             The conclusion has been modified.

  1. The future prospects should contain some application of the EDM.

             Relevant applications of EDM has been added.

Reviewer 2 Report

The current article pertains to an interesting topic, namely, a more theoretical approach of the micro-EDM process is attempted. Definitely authors have deep knowledge and understanding of the relevant subject, and hence, they present a robust theoretical methodology guideline and suggestions for future research. Nevertheless, and in the scope of the scientific debate, there are some points that have to be clarified.

More specifically:

  • authors state: "It is assumed that the volume of the material removed from the workpiece and from the tool-electrode is proportional to the discharge energy...". It is well known that the volume of the removed material is not proportional to the discharge energy in macro-EDM, hence, a clarification and distinction between micro-EDM and macro-EDM is necessary here. Authors probably are referred in micro-EDM, and this has to be clearly stated.
  • a consistent nomenclature is missing. For example in eq. 6 the terms are not adequately explained.
  • since the current paper is mainly based on experiments, results and conclusions of a previous study (Ferri, C., Ivanov, A. and Petrelli, A., 2008. Electrical measurements in μ-EDM. Journal of Micromechanics and Microengineering, 18(8), p.085007) a little more detailed presentations and reference on this work has to be included, in order the current paper to not be considered as complimentary to a previous work.

The current paper is a perspective article, and thus it can mainly be considered as guideline for future research. This justifies the brief analysis and discussion, the preliminary results, as well the extended "Future Prospects" section. As an overall conclusion, and keeping in mind the nature of a perspective article, it can be accepted after a minor revision.

Author Response

. authors state: "It is assumed that the volume of the material removed from the workpiece and from the tool-electrode is proportional to the discharge energy...". It is well known that the volume of the removed material is not proportional to the discharge energy in macro-EDM, hence, a clarification and distinction between micro-EDM and macro-EDM is necessary here. Authors probably are referred in micro-EDM, and this has to be clearly stated.

Clarification is been added in the paragraph.

· a consistent nomenclature is missing. For example in eq. 6 the terms are not adequately explained.

All the nomenclatures have been looked into and included in the manuscript.

· since the current paper is mainly based on experiments, results and conclusions of a previous study (Ferri, C., Ivanov, A. and Petrelli, A., 2008. Electrical measurements in μ-EDM. Journal of Micromechanics and Microengineering, 18(8), p.085007) a little more detailed presentations and reference on this work has to be included, in order the current paper to not be considered as complimentary to a previous work.

Additional references have been included in the manuscript.

The current paper is a perspective article, and thus it can mainly be considered as guideline for future research. This justifies the brief analysis and discussion, the preliminary results, as well the extended "Future Prospects" section. As an overall conclusion, and keeping in mind the nature of a perspective article, it can be accepted after a minor revision.

Round 2

Reviewer 1 Report

The paper is well done in the present form. There are only two things need to be improved.

  1. The unit shown in Y axis in figure 4 is not clear.
  2. The format (including the font) needs to be uniform.